# Wildfire and Smoke Risk Communication: A Systematic Literature Review from a Health Equity Focus

**DOI:** 10.3390/ijerph22030368

**Published:** 2025-03-03

**Authors:** Sofia Sandoval, Jessica Bui, Suellen Hopfer

**Affiliations:** Department of Health, Society & Behavior, School of Population & Public Health, University of California, Irvine, CA 92797, USA; sofias6@uci.edu (S.S.); jbui4@uci.edu (J.B.)

**Keywords:** wildfire risk communication, smoke risk communication, public health, communication

## Abstract

Effective wildfire smoke risk and evacuation communication is urgently needed to avert unnecessary deaths as wildfires increase in frequency and intensity. Human exposure to wildfires has doubled in the last two decades. Low-income, marginalized communities are the most disadvantaged in their ability to respond. A systematic literature review of wildfire and wildfire smoke risk communication research between 2014 and 2024 was conducted. Web of Science and Scopus databases were searched using the keywords “wildfire”, “communication”, “wildfire smoke”, “risk”, and “public health”, resulting in 23 studies. The findings revealed marginalized communities were ill-prepared to respond to wildfires and take protective action against wildfire smoke. The findings were summarized across eight areas: the needs of marginalized communities to respond to wildfires, the role of trusted messengers to disseminate wildfire and smoke risk messaging, using diverse channels, timing and frequency considerations for disseminating messages, time-sensitive evacuation versus wildfire smoke risk messaging, targeted messaging for subgroups, the importance of coordinating messages across agencies and local government, and government perspective. Theory did not guide these research efforts with the exception of one study, and most studies were qualitative. The literature did not report on distinguishing indoor from outdoor protective action against smoke risk, reaching vulnerable communities such as nursing and older adult facilities, and postfire messaging. Evidence is needed on these fronts, along with experimental messaging studies to determine the most persuasive messages for motivating protective actions against wildfire and smoke risk.

## 1. Introduction

In 2024 alone, 55,571 wildfires happened globally, releasing 2170 megatons of carbon emissions into the atmosphere [1,2,3]. As climate change progresses, new conditions of combined drought, heat, and wind have created ideal environments for wildfires to grow larger and longer [4,5,6,7]. The culmination of these factors has led to the secondary effect of severe smoke conditions both during and after wildfires. Wildfire smoke is a growing driver of air pollution, erasing improvements in air quality and heavily affecting health, given it may be more toxic than other air pollutants depending on its composition and concentration [8]. For this review and according to the West Fire Chief Association, we define wildfire as an unplanned and uncontrolled fire occurring in wildlands such as forests, rangelands, or grasslands. The term wildfire encompasses three fire types: ground or subsurface fires; surface fires; and crown fires, which burn in the tree canopy [9,10]. Smoke events from wildfire more often than not cause more death and injury than the initial fire.

A growing body of epidemiological evidence suggests that exposure to wildfire smoke, particularly fine particulate matter, is associated with health effects on multiple systems, including the respiratory, ocular, cardiovascular, dermatological, nervous, immune, digestive, and reproductive systems. Toxicological studies have shown that inhaled wildfire particulate matter deposited in the lungs and entering the bloodstream can cause intense systematic inflammation and oxidative stress, leading to pulmonary and vascular dysfunction and contributing to the development of cardiorespiratory diseases. Inhaled particulates also may disrupt the autonomic nervous system, which has been linked to cardiovascular health effects across gender, age, race, and region, with children and older adults particularly vulnerable to these adverse effects [11]. Acute health effects such as asthma and chronic obstructive pulmonary disease and increased risk of hospitalizations and emergency department visits for these conditions have also been documented [12,13,14,15]. Mental health effects (e.g., anxiety and depression) also result from inhaled wildfire particulate matter and wildfire experiences. Although research on the long-term effects of wildfire smoke on health is limited, it is known to affect health across the lifespan with repeated exposure (e.g., among fire fighters), leading to worsened outcomes including cancer [9,10,16].

Research on effective methods to communicate wildfire and wildfire smoke risks, both health and emergency related, is lacking. It is especially not well understood among marginalized communities, such as Indigenous, farm worker, low-income, rural, and older adult populations [17,18]. These groups are in some cases geographically isolated and reside in unincorporated communities (in the United States), which have little to no local governance structures to assist with wildfire and smoke mitigation. These communities may also exist in linguistic isolation (e.g., Spanish only), have low knowledge about what to do in the event of a wildfire and evacuation, and experience greater exposure to general air pollution as a consequence of environment and occupation [18,19]. Other marginalized communities include low-income, older, disabled, or low-education groups that are disadvantaged and may reside in urban–wildland interface or rural areas [20]. Consequently, to save lives, greater awareness and outreach are imperative to reduce harm from acute wildfire and minimize potentially severe long-term harmful health effects from smoke exposure. For outreach and education efforts, determining trusted sources and using multimodal communication to convey wildfire and smoke risks are also critical [21].

This systematic literature review sought to identify gaps in studies about wildfire and wildfire smoke risk communication. This review also pinpointed qualities of effective communication for acute evacuation versus repeat smoke exposures and identified strategies for marginalized communities. With an aim of promoting health, this review generated recommendations to formulate a research agenda on risk communication, particularly wildfire smoke risk communication.

## 2. Methods

### 2.1. Study Inclusion Criteria

We excluded noncommunication, climate change, and health-focused literature and gray literature. Additionally, any non-English papers were excluded from the initial search. Epidemiological studies on adverse health effects associated with wildfire smoke exposure with little to no focus on smoke messaging were excluded. Social media studies on monitoring wildfire conditions, studies about prescribed fires, and studies focusing on co-exposure were excluded. We included the literature that focused on the intersection of communication methods and active wildfire and smoke event information or protective messaging. The included articles provided information on types of messages and channels that were effective in mitigating the adverse effects of wildfire or smoke and advancing communication. Additionally, the inclusion criteria included providing explicit recommendations about communication efforts. The study types included qualitative studies involving interviews with community members; input from experts at workshops, conferences, or symposiums; or input from the focal population. International papers were included as long as the language was English.

### 2.2. Search Criteria

In the initial search round, the keywords entered into the engines were “wildfire” AND “communication” AND “public health”. In the second search round, the keywords entered into the engines were “wildfire” AND “communication” AND “public health” AND “risk” AND “smoke” OR “wildfire” AND “communication” AND “public health” AND “risk” AND “wildfire smoke.” The first round of searches provided a greater selection of articles (*n* = 109), whereas the second round of searches was more refined (*n* = 36; see PRISMA flowchart in Figure 1). Both searches were conducted to ensure comprehensive identification of all relevant articles. The search scope was limited to the last 10 years (2014–2024) to keep information relevant because the circumstances of wildfire events are consistently evolving with climate change and technological advances. The databases selected for this review were Web of Science and Scopus. The Sociological Abstracts database was also searched but ultimately excluded because it did not yield additional unique studies. Both the Web of Science and Scopus databases are interdisciplinary. Scopus and Web of Science cover the intersection of wildfire, smoke, communication, and health and a breadth of biomedical literature, including publications focused on public health and content from across the globe. Scopus has a fast process for indexing new papers, which allows for analysis of the most up-to-date information on wildfire and smoke communication. The databases were last searched in July 2024.

### 2.3. Study Eligibility

Abstracts were reviewed to ensure studies aligned with the inclusion criteria: English, international, published in the last 10 years (2014–2024), and focused on communication. Studies that did not focus on communication were excluded, as were articles that only examined health outcomes. If a study met the inclusion criteria, it was added to a spreadsheet for further annotation. The spreadsheet allowed for record keeping of titles, authors, dates, and notes on each study. Two research team members (authors S.S. and J.B.) completed this process independently for both databases, Web of Science and Scopus. Reviewers worked independently to search each article and categorize its content into the spreadsheet. No automated tools were used.

Deciding on the final selection of headings and presentation of findings (by all three authors) involved a five-step process: We (a) prioritized communication-relevant variables and considerations for wildfire and smoke; (b) highlighted emergent findings from our 23 mostly qualitative studies; (c) considered crisis communication theory (crisis messaging for short and long terms and stages) and the warning response model, a psychological model that emphasizes shifting perceptions of safety to perceptions of risk; (d) deliberated on how to present the findings in the most parsimonious way (e.g., we initially had a column on messaging before, during, and after a fire but changed this to wildfire smoke vs. acute evacuation messaging); and (e) focused on marginalized community needs as a distinct point of our review. After synthesizing key summaries, we had eight section headings for our findings: marginalized communities, trusted messengers and sources, channels, timing and frequency, wildfire smoke risk vs. evacuation messaging, targeted messaging for subgroups, coordination between agencies, and government perspectives [23].

In reviewing the spreadsheet, any studies determined to fall outside our scope were removed. For articles with ambiguous content, we discussed their relevance. If the article was determined not to provide additional value, it was excluded from the analysis. Table 1 and Table 2 were created to organize the papers: (a) a list of the final 23 studies with year, author, location, sample size, and type of study and (b) abbreviated findings and level of evidence through designation of randomized trial status. In Table 2, one outcome was whether the articles mentioned evacuation or smoke protection messaging. Evacuation messaging is used during time-sensitive emergencies, whereas smoke protection messaging can be used year-round and focuses on reducing exposure. Another outcome was whether the studies focused on communication for individuals, such as tailoring messages to meet individual needs, or communities, which are messages addressing wider audiences. We listed the trusted sources and channels preferred by each population. Trusted sources are organizations, authoritative figures, or community leaders whose messages are received and trusted by populations. Channels are modes of communication by which people receive information about wildfire smoke, such as social media, newspapers, or radio. Finally, we retrieved the recommendations and lessons learned proposed by the authors or populations in each study. One included study involved a randomized messaging experiment, whereas the remaining studies were based on qualitative designs or information from discussions at symposiums or workshops. The selection process was constrained by the outcomes available in the databases.

## 3. Results

Of the 109 articles initially identified, 23 studies were deemed to have focused on wildfire risk communication. The review covered the last 10 years (2014–2024). Of the final 23 studies, three (13%) were conducted in Canada, five (22%) in Australia, and 15 (65%) in the United States.

### 3.1. Marginalized Communities

Ten of the 23 studies (43.5%) discussed effective wildfire smoke risk communication among or for marginalized communities. The communities mentioned encompassed Indigenous, rural, immigrant, and certain occupational groups. Employing diverse channels of communication ensured greater reach in marginalized communities. Indigenous people, certain occupational groups such as those in trades, and rural areas may be geographically constrained, with limited internet and cellular connection [26]. Beyond geographic limitations, studies focused on variations in risk perceptions depending on the culture and local environments of different communities. Traditional modes of messaging, such as radio, newspapers, mail, word-of-mouth, and mass phone calls, were preferred by marginalized communities [26,31,37,38,46]. VanderMolen and colleagues [36] found that rural communities in Nevada almost exclusively preferred local communication channels, with the greatest preferences for county websites, social media, and community Facebook pages.

In addition, translating messages to languages other than English helped engage Indigenous and immigrant populations. Translating social media posts and contacting non-English radio and television stations were ways to increase the accessibility of communication [26,30]. Walsh et al. [33] highlighted that in Eastern Australia, providing information in languages other than English was beneficial even for English-speaking parents and caregivers of children because it offered parents reassurance when communicating health information in a family setting. Translation also benefited social groupings in which parents who spoke English as a second language supported their extended same-language family.

A recommendation for improving risk communication with Indigenous communities was to engage trusted individuals or groups, such as local and tribal firefighters, in disseminating messages to promote protective behaviors [24]. In tribal communities of the Okanogan River area in Washington, trusted channels took the form of informal networks, such as friends, family, and other community members, and private, crowd-sourced, citizen-run fire watch Facebook pages [31]. D’Evelyn et al. [24] emphasized the importance of coordinating consistent and accurate information among local tribal groups in rural Washington. Emphasizing community health also encouraged wildfire resilience. Ensuring successful wildfire and smoke protection messaging in tribal communities requires investment in year-round smoke readiness education and training [24]. Wood et al. [31] recommended that centering local perspectives and expertise in designing risk communication can be a strategy to target messages for subgroups.

### 3.2. Trusted Messengers and Sources

Ten studies (44%) surveyed and interviewed residents in high-risk wildfire smoke exposure areas. Households exposed to wildfires in Bastrop County, Texas, in 2011 reported a higher likelihood of trusting local government responses to wildfires. Past response and recovery efforts initiated by the local government bolstered this trust [27]. In the Huon Valley region of Tasmania, Australia, people expressed greater trust in smoke-related advisories disseminated by government agencies or community members than social media. Social media was perceived as containing misinformation and overblown reactions to smoke risk. On the other hand, information from government agencies and trusted community members was perceived as more reliable and valued [34]. In Mariposa County, California, trusted sources included the Mariposa County’s sheriff office, county health agency, community, and local fire departments [38]. Rural residents of Washington identified local social or health service providers as trusted sources [42]. People with asthma or chronic obstructive pulmonary disease in an Australian community suggested information should be communicated by websites or social media groups such as Asthma Australia or the Lung Foundation. In addition, the source of information should be disclosed, such as the location of monitoring stations, and wearing masks should be encouraged by healthcare providers during bushfire smoke events to increase trust and use of masks [41].

Two studies [24,31] assessed how tribal and nontribal residents of the Washington Okanogan River community interacted with smoke risk information and sources. Tribal communities expressed greater trust in tribal government, local sources, informal networks, and nongovernmental and community-based organizations than state and federal government sources. Study findings revealed that state and federal governments were less trusted by communities due to a perceived disconnection with local issues, political agendas, lack of accountability, and negative past experiences. Tribal government sources, such as information from Confederated Tribes of the Colville Reservation agencies, were perceived as locally and culturally relevant. Local sources included conservation and fire departments, sheriffs, emergency management agencies, schools, and public utility departments. Trust in local sources was attributed to perceived neutrality, authority on the topic of fires, and presence in communities [24,31]. Shared values and lack of conflict on worldviews influenced trust among informal networks. Private citizen-run Facebook pages were another trusted source because information was seen as timely, locally relevant, and crowd-sourced [31]. Trustworthiness was determined by various factors. Tribal members identified authority and expertise, accuracy, timeliness, local and tribal relevance, and relationships as the most important factors. Nontribal members identified perceived political neutrality, transparency, authority, and respect as the most important factors contributing to trust [24,31].

Four articles [25,31,35,46] studied trusted sources from the perspective of institutions and officials who disseminate smoke risk messages. In a study that analyzed the content of wildfire smoke information communicated by government organizations and mainstream media in Washington, Van Deventer et al. [25] defined trusted messages as those that mention a government agency or academic organization. Results from the study revealed that trusted sources referenced by government and media messages varied. Government messages mostly included sources like clean air agencies, Washington State Department of Health, and Washington State Department of Ecology, with little mention of medical care facilities. Media messages mostly cited clean air agencies and the health department. Across studies, risk messages with air quality descriptions failed to specify whether the reference was the Environmental Protection Agency’s Air Quality Index (AQI; federal) or Washington Air Quality Advisories (state), which differ in air quality measures [25]. Aminpour et al. [32] found that Facebook users had no preference between academic (i.e., “.edu”) and government (i.e., “gov”) sources. Treves et al. [30] explored trusted sources of practitioners in unincorporated Northern California who activate clean air centers (CACs), which are public buildings that provide adequate air filtration to communities during wildfire smoke events. Practitioners said outreach partnerships were most able to mobilize and rapidly reach populations vulnerable to smoke. Outreach partners included nursing facilities, human services agencies, city-contracted service providers, nonprofit nongovernmental organizations, and social media influencers.

To establish trustworthiness, risk messages should include both risk and efficacy messages to increase message acceptance and adoption of recommended protective health behaviors. State and local departments should increase their presence as trusted sources during statewide smoke events. Local health agencies are effective trusted sources of communication because they are most familiar with their community’s vulnerabilities, capacities, and cultures [25,31].

When communicating with rural, tribal, and unincorporated communities, state and federal agency practitioners should demonstrate accountability to communities and acknowledge historical harms and disinvestment. Furthermore, government agencies must cultivate and build long-term relationships with communities to increase perceived credibility and trust. This is particularly challenging to do in unincorporated communities that lack local government. Tribal nations should be included in state and federal decision-making and risk communication design. Government agencies should also support local and tribal partners in smoke readiness, employment of community members, and development of resources by providing funding and grants [31].

### 3.3. Channels

Many included studies examined communication channels (83%), highlighting the importance of planning diverse dissemination strategies for effective smoke risk communication. However, interviewees in two studies (one in Canada and one in Tasmania, Australia, surveying residents after a recent wildfire smoke event) found having many sources to be burdensome. The study respondents advocated a central resource with links to different information about wildfire smoke [26,34]. British Columbia residents accessed wildfire smoke information via four channels: websites, social media, radio, and television, with radio being the most important for Indigenous communities. Many residents surveyed (60%) were unaware of the ability to sign up for automated alerts about wildfire smoke. Among Tasmanian residents after a 2019 wildfire, social media played a major role in receiving timely wildfire smoke information. These residents stated that multiple sources of information caused confusion and gave conflicting messaging. The choice of channels depended on various factors. First, channel preferences depended on the population. Tribal communities preferred tribal broadcast emails, community information boards, local and tribal news, and weather apps as modes of communication [24]. Nontribal communities used less centralized sources, including community-based organizations, local news and media, and local government agencies like conservation districts, schools, and fire districts [24]. Both tribal and nontribal populations used websites, social media, radio, television, and informal networks like friends and family [24,26]. In the Okanogan River basin of Washington, which features both tribal and nontribal residents, Facebook was widely used by the Confederated Tribes of the Colville Reservation, local government, community agencies, and residents [24,31]. Last, in a Canadian study (British Columbia), residents cited that they turned to websites, social media, radio, and television for wildfire smoke information, with websites being the most frequent source [26].

Channel preferences were also determined by location. For instance, rural and Indigenous populations relied on traditional channels like radio, newspapers, mail, the library, and informal networks, along with websites and social media [26,31,36,38]. Other sources referenced by rural communities in Nevada included senior centers, billboards, flyers, fire departments, pamphlets in medical offices, AirNow.gov (federal website), community leadership, text, and the local market [36,46]. In Traralgon, a large regional center, people received information from a range of channels, including radio, social media, internet sites, and bushfire alert apps. However, residents of Traralgon were less likely to seek wildfire smoke information [46].

Preference for channels also changed depending on the type of message that was being disseminated. Respondents in Shellington et al.’s [26] study, which enlisted patients with chronic obstructive pulmonary disease and asthma together with local air quality council members and Canadian residents, recommended that designated air quality reports be disseminated via news outlets. Households in Bastrop County, Texas, preferred television, social media, word of mouth, newspapers, and radio for local news and information. For emergency information from the local government, people selected text message, television, social media, radio, and email. When being informed about emergency information, respondents preferred the U.S. Postal Service, email, and social media [27]. People’s choice of channels varied depending on whether the information focused on the past or the future. Past-focused information was often sourced from local news, whereas future-focused information was retrieved from government sources [38].

Two studies recommended that agencies increase the use of social media and digital messaging because these platforms enable content to be easily shared through online community networks [27,31]. In Treves et al.’s study [30], community members emphasized using multiple communication methods to learn about activating CACs, public buildings designated as safe places for wildfire smoke events. The communication methods included flyers; schools; radio; Spanish language newspapers and television; and social media, such as Nextdoor. Community members also mentioned the use of interactive methods of communication, including charlas, defined as an informal talk or discussion, and direct messages from health clinicians [30].

In the case of communicating to children, some parents said television, radio, news, or social media were not appropriate channels. Meanwhile, some parents found internet video sites to be useful because they displayed practical smoke protective steps. Parents typically served as the primary source of information for children. For instance, parents shared information on air quality apps with their children because the simple language and color coding were digestible [33]. People with asthma or chronic obstructive pulmonary disease in Australia expressed mixed reviews about air quality apps, citing system errors, lack of user-friendliness, and distrust in the information due to data collection points being inaccurate for their locations. Other relied on websites from the Department of Environment and apps. The high-risk community expressed preference in using text alerts for air quality and health protective behaviors and reminders about bushfire precautions through Royal Fire services, radio, council or government notices, news, and local social media networks focused on health conditions. The authors recommended the use of auto-alert systems via telephone [41]. Meanwhile, rural residents of Washington recommend a range of mediums, including printed materials like brochures, online materials like websites and social media, radio, and a hotline [42]. Outdoor workers in Colorado mainly cited local news and media outlets as their primary channels for air quality information, followed by the Weather Channel, apps, state public health authorities, the National Weather Service, websites like Purple Air and EPA Air Now, friends and family, social media, healthcare providers, local governments, and other channels [43].

### 3.4. Timing and Frequency

Four studies (17%) analyzed the role of timing and frequency of disseminating smoke messages and how they affect the response and access to evacuation and smoke information. A common recommendation across studies was that timing varies across populations. Factors like location and age group affect how often fire and smoke messaging needs to be disseminated [46]. A similar outcome was seen by Hoshiko et al. [38], such that members of a rural Californian community affected by fires stated that the smoke events created power outages that made receiving emergency messaging impossible. Similarly, they called for faster communication so critical information would reach rural communities before power was lost.

Communities affected by fire and smoke events voiced the need for preemptive messaging at the beginning of and throughout the fire season [26]. By providing updated health protection information, families in rural communities would feel more prepared for high-risk fire seasons. Taking preparatory steps before the fire season initiates could mitigate the consequences of poor health messaging that occur during a fire and smoke event [26]. Messages should be simple while also displaying clear guidelines [34,46]. Hoshiko et al.’s study [38] in Mariposa County also found that older individuals required regular reminders outside of fire season to keep their medical equipment, such as respirators, ready for an emergency evacuation, given these devices require power.

### 3.5. Evacuation vs. Wildfire Smoke Messaging

Twelve studies (52%) focused on evacuation messaging, and all 23 studies focused on wildfire smoke risk messaging. Although evacuation and smoke communication had some overlap, evacuation messaging was time-sensitive, whereas smoke risk messaging focused on reducing exposure to adverse wildfire smoke. Each type of communication—evacuation and smoke risk messaging—involves different information and goals.

Evacuation messaging needs to be created and disseminated in a time-sensitive manner. Additionally, messaging may need to change over time in a dynamic fire situation, based on factors like wind conditions and development of the fire (e.g., evacuate as quickly as possible vs. shelter in place) [25]. This creates complications in evacuation messaging because widespread communication may be limited by the fire (e.g., power outages). Additionally, evacuation messaging also needs to account for cultural and age differences. Using trusted sources and diverse communication channels may reduce barriers to reaching affected communities.

During an evacuation event, scholars found that it is best to avoid conflating smoke and evacuation information. Marfori et al.’s study [34] of Australian smoke and evacuation broadcasting showed people felt that widespread smoke information “drowned out” lifesaving evacuation notices. Smoke messaging, unlike evacuation messaging, could be produced and shared year-round. Smoke communication tended to emphasize preparation and health protection, whereas evacuation messaging focused on exit strategies. State and local departments should coordinate with interagency partners to distribute smoke protection and evacuation messages. These local entities are most familiar with the cultures, vulnerabilities, and capacities of their communities [25,27]. Increasing year-round social media campaigning with smoke protection messages to improve preparedness was recommended.

Shellington et al. [26], Slavik et al. [35], and Dodd et al. [45] advocated both smoke protection and evacuation messages. These researchers found that it is important to keep a consistent smoke messaging agenda before, during, and after smoke events [24,25]. D’Evelyn and colleagues [24] recommended that smoke protection messages be targeted to an organization or household. Dodd and colleagues [45] highlighted the importance of smoke forecasting on physical and mental health and recommended debriefing after a wildfire season to improve intervention strategies.

As for content, promotion of health protective behaviors was viewed as critical for mitigating health repercussions during heavy smoke. For instance, advocacy regarding air filter installation and maintenance helped communities create clean air rooms in their homes [25]. Gaps in health promotion in smoke messaging consisted of the lack of AQI information, including what the AQI scale means and how a poor AQI indicates health risks for sensitive groups [28].

### 3.6. Targeted Message Content for Subgroups

Authors of eight studies (35%) recommended that risk messages include information about the health impacts of wildfire smoke on individuals and communities [24,25,28,41,42,43,44,46] and actionable steps to protect health. In particular, participants from Marfori and colleagues’ [34] study recommended that health risk messages emphasize long-term effects, present risks to all populations including those not categorized in the higher-risk group, and clarify the risks associated with different levels of smoke pollution severity. In addition, Rappold et al. [28] recommended that the focus shift toward messages about health risks rather than air quality information as a strategy to increase health protective behaviors. Rappold et al. also recommended that risk messages include evidence highlighting the effectiveness of protective health behavior.

Messages may need to be targeted to different populations, especially for marginalized populations [24,25,46]. As previously noted, protective actions or health risk messages may need to be highlighted or provided in another language to reach subgroups effectively, especially marginalized communities [28,34]. When clarifying action steps to mitigate or reduce people’s smoke exposure, protective actions need to be clearly and concisely communicated. In some cases, using a trusted messenger will be a key consideration, given the historical interactions between a community and government officials [24,46]. In addition to targeting wildfire risk communication by language, other considerations may include focusing on families with members who have disabilities and need extra time to evacuate. Australian residents with asthma or chronic obstructive pulmonary disease recommended that organizations provide a rationale and instructions for wearing a mask when exposed to smoke [41]. In rural Washington, residents suggested message content should consist of health and social resources via links and hotlines, communication training for healthcare workers, and stress reduction methods [42].

One way to increase the reach of messages is to shift from a fact-focused informational framing to a story-focused narrative framing [32]. Another strategy is to combine numeric information, verbal cues, and AQI risk labels when communicating about wildfire smoke. In addition, engaging and bidirectional communication is needed, whereby messages emphasize both community health and environmental protection [35]. Slavik et al. [35] also recommend that institutional messages integrate protection motivation theory to increase action. Protection motivation theory is a social psychology model positing four key constructs in motivating protective health behavior: risk severity perception, likelihood of experiencing harm, effectiveness of mitigative measures to protect, and belief (i.e., self-efficacy) that one can execute the mitigative measures.

Walsh et al. [33] investigated the communication needs of children and their caregivers during smoke events. When developing communication for children, it is important to consider age and developmental level. A child’s level of social and cognitive development influences how they perceive a threat. For instance, younger children carry a more imaginative mindset, whereas older children use information to guide their behavior.

When messaging outdoor workers, it is important to highlight the effectiveness of health-protective actions to motivate information-seeking behaviors. Communication should be personalized and actions should be practical for outdoor workers, in alignment with workplace policies and norms. A gap remains between behavior recommendations about air quality to the public versus outdoor workers. For instance, remaining indoors during periods of dangerous air quality is not feasible for this audience [43].

### 3.7. Coordination Between Agencies

Coordination among local groups, government, and news media is necessary to ensure accuracy, consistency, and reach of wildfire and smoke risk communication, as noted by five studies (22%). These entities can coordinate to develop a centralized source of information and formalize the process of sharing messages from other agencies to ensure the dissemination of accurate and consistent information [24,25,34]. D’Evelyn and colleagues [24] suggested using a risk communication framework designed for local groups and agencies to develop risk communication plans. The information developed from coordination may encompass the health risks of smoke exposure, strategies to reduce exposure, and air quality indexes. For example, consistent messages about AQI could reduce confusion about the air quality and associated health risks. Errett and colleagues found that different agencies such as the Environmental Protection Agency and Washington Department of Ecology had differing definitions of “unhealthy” and “unsafe” [40]. Coordinated content may include risk communication tools, templates, and resources. For example, state and local health departments, clean air agencies, and other stakeholders can coordinate to create a risk communication campaign in preparation for wildfire smoke seasons [25]. In rural communities, information campaigns about available resources, air quality forecasts, and areas with clean air should be implemented. Physical signage that interprets AQI values, multiple formats of communication including bulletin boards and roadside signage, and push notifications on local apps for harmful AQI levels were also recommended by rural communities [42]. Two articles recommended coordination to create a single source with easily accessible links [24,34]. An accessible and centralized resource would improve messaging, self-efficacy, and health [34].

### 3.8. Government Practitioner Perspectives

In Treves and colleagues’ study [30], practitioners activated a CAC by identifying a large public building with adequate filtration during a smoke event. To improve the reach of information about the CAC to local communities, practitioners recommended clear guidelines and community engagement to identify familiar and safe locations. The practitioners also recommend tools to plan public transit, greater allocation of resources to meet staffing needs, and tools and funding to share information about CACs and wildfire smoke preparedness proactively. Humphreys et al. [42] found that local health and social service providers in rural Washington viewed the quality and quantity of public health communication during wildfire seasons more positively than residents did. This difference in perspectives may stem from practitioners possessing greater wildfire smoke literacy due to their professional expertise compared with the general public.

Practitioners and academics in Washington acknowledged the need for messages relevant to subpopulations. Research is needed to understand the motives of different subpopulations, how to communicate risk to dismissive groups effectively, the impact of using analogies such as comparing wildfire smoke with cigarette smoke, and the effectiveness of emphasizing health outcomes or specific interventions. Practitioners and academics recommended using community-level asset mapping to determine the trusted messengers with which subpopulations resonate [40]. Bushfire experts and representatives of groups that have been affected by bushfires supported the use of hourly reports for pollutant concentrations and air quality because they are more reflective of acute exposure. Challenges include the complex underlying characteristics of each individual and the problem of deciding whether to generalize messages or target subpopulations. The experts recommended that messages avoid panic and encourage health-protecting actions. Messages for healthy individuals should motivate protective actions by emphasizing the harms of exposure over time. Panelists from a wildfire symposium called attention to the need for improved environmental health literacy through educating and engaging with communities, possibly using campaigns. Concerns included a media focus on crisis communication, politicization of climate change, and inadequate reporting of health impacts, along with a need for storytelling in journalism and evidence translation for policy [44].

## 4. Discussion

In our systematic review across 10 years (2014–2024), we identified 23 studies concerning wildfire and wildfire smoke risk communication. Three types of studies were identified: mixed-methods qualitative interview and survey studies with communities (*n* = 18), website wildfire risk content reviews (*n* = 2), and smoke app reviews (*n* = 3). Studies on wildfire communication in the United States were predominantly conducted in western and southwestern states (California, Washington, Nevada, and Texas), whereas international studies reported on wildfire risk communication in Canada and Australia.

Among the 23 studies, slightly more than half covered risk perceptions and needs of marginalized communities (e.g., tribal communities and low-income, non-English-speaking immigrant and farmworker communities) that are disadvantaged at many levels in responding to wildfire threats. Given our health equity lens in this review, key factors emerged as relevant for bolstering marginalized communities’ preparedness: ensuring greater local government interaction with isolated (often unincorporated) and marginalized communities and offering disaster preparedness in languages other than English and in smaller unincorporated communities [18].

Another emergent theme in more than half of the studies was the recognition and importance of using trusted messengers when communicating wildfire evacuation or wildfire smoke risk messaging. Trusted messengers are in many cases local leaders. Local leaders had more influence with respect to compliance with evacuation messaging. Later evacuation has been associated with more fatalities [47]. Mobilizing community residents to evacuate promptly when a fire threat is imminent is critical. Studies covered evacuation versus wildfire smoke protection messaging, the importance of trusted messengers in disseminating wildfire risk communication, using diverse and nimble channels, timing and frequency of messaging, and emphasizing health impacts and actionable steps.

Diversity of channels was recognized as important across all studies. Context and community factors should ultimately determine which channels to use to communicate wildfire risk with a particular community. Local government agencies should budget to diversify and use multiple channels. Advantages of electronic alerts and social media are that they are responsive and have wide reach, which are important for evacuation contexts. However, to reach isolated, rural communities and certain occupations that do not access electronic alerts readily and when power outages occur, nonelectronic wildfire communication must also be in place (including sirens, radio, television, websites, road signs, and red flag alerts). Diverse channels should be planned regardless of the community, as recognized by Fish et al. [21] and Hopfer et al. [18,48]. Traditional, nonelectronic channels (radio, television, and sirens) are better for older populations or those working in occupations who do not routinely access their phones, social media group chats, locally familiar websites, or tribal email listservs. Awareness about the channels used by marginalized, isolated, and rural communities needs to be prioritized in the planning of wildfire communication. Communicating fire risk, evacuation, and smoke warnings in other languages was recognized as important in in multiple studies [18].

An advantage of social media or electronic alerts is their ability to respond rapidly to fast-changing fire conditions for evacuation. Smoke messaging, on the other hand, which may be repeated, year-round, and related to prescribed burns, can focus on protection from health risks. Smoke messaging should be coordinated by local government agencies, including local air quality districts, fire departments, and county sheriff offices. Air quality and fire departments may not communicate with each other on a regular basis. Informal and tribal government networks communicate wildfire and smoke risk reduction education across local networks and organizations (e.g., schools and senior centers). Absent from the review were studies distinguishing health protective actions and behaviors for indoor versus outdoor environments. This is an area in which research evidence is needed. Also, although diverse channels are necessary, having communities be familiar with one centralized source is equally important. For instance, among tribal networks, tribal broadcast emails, community information boards, local and tribal news, and weather apps are acceptable for receiving wildfire communication.

Missing from the review of 23 studies was the use of theory to learn what key features are essential for accurate wildfire risk perception, response, and compliance with recommended protective behaviors. Two studies [18,35] referenced theory. One study considered protection motivation theory when designing wildfire smoke risk messaging that institutions disseminate. Disaster and risk communication theories like the protective action decision model may be relevant for wildfire evacuation and smoke risk messaging. This multistage model is grounded in people’s response to environmental hazards and focuses on reception, attention, and comprehension as necessary precursors to compliance and persuasion relative to fire evacuation messaging or smoke exposure reduction goals [47,49]. In one study, the response efficacy was the only predictor of emergency preparedness [50]. Emphasizing response efficacy and expected personal consequences in protective action messages are key constructs to motivate behavior change. Sutton and colleagues [51,52], whose research focused on designing effective warning messages for wireless emergency alerts, used Mileti and Sorenson’s warning response model to guide communication efforts [52,53,54]. This model focuses on including key content features for behavior compliance, including the source, naming the hazard explicitly, hazard impacts, the location, the protective action requested, and potentially links to more information. The other study from the review [18] used a wildfire community vulnerability framework focusing on five areas: personal lived experience with wildfire, health impacts, response and mitigation behaviors, community social interaction, and wildfire risk communication preferences.

Nearly half of the reviewed studies focused on wildfire risk communication for acute, time-sensitive evacuation or smoke messaging. Distinguishing evacuation from smoke messaging can reduce confusion. Evacuation messaging is time-sensitive and focused on directing residents to expedient evacuation routes, alternative routes, life-saving actions, and shelters. By contrast, smoke messaging may be needed for a longer duration to emphasize preventive measures and distinguish indoor from outdoor protective actions. Also, smoke messaging can be year-round and familiarize communities with protecting themselves against health risks from repeat smoke exposure [26,55]. Branding efforts could render smoke protection messaging more memorable for communities to readily mobilize and enhance trust.

Although one study mentioned the importance of indoor filtration systems to protect against adverse health risks, distinguishing indoor from outdoor protective action steps regarding smoke was largely absent in the literature. A systematic review of short-term exposure to wildfire particulate matter on respiratory outcomes demonstrated the importance of protection, especially children’s exposure to particulate matter in indoor environments [15,56]. This constitutes a future research agenda to generate evidence for indoor protective actions.

Coordination among local agencies was also recognized as a critical structural factor to increase effectiveness in wildfire risk communication. Two studies [18,25,31] recognized the importance of interagency coordination among local fire agencies, air quality districts, health departments, and emergency response agencies to disseminate wildfire risk messaging effectively. The role of interagency coordination has long been recognized in the risk communication literature [57] for effectively mobilizing communities, increasing trust, and having consistent risk messaging across agencies. Interagency coordination during the COVID-19 pandemic might offer a model for future efforts.

Federal and state smoke apps have been developed to aid community residents with risk assessment of wildfire smoke and air pollution effects. However, few residents use these smoke apps. Our review revealed three studies captured individual traits that motivated seeking air quality information. Few residents, unless personally motivated because of prior wildfire experience, are familiar with federal or state smoke apps [58]. Researchers from UCLA designed a smoke app to make air pollution risks visible to users. This study examined engagement in using and sharing air pollution app information [58].

Limited evidence exists to know under acute wildfire evacuation conditions when people process messages differently what messaging is critical. For evacuation messaging, Sutton and colleagues extensively researched message features that may lead to greater behavior changes and sharing of protective health information under conditions of stress and imminent threat to life [52,53].

### Future Directions

Targeting the structural determinants of wildfire and smoke risk mitigation should be prioritized, especially for marginalized communities. For example, for unincorporated isolated communities, greater initiatives are needed on the part of local governments at the county level to reach out and provide wildfire and protective action education to marginalized communities [18].

Marfori et al. [34] recommended more research to improve evidence on the range of interventions for smoke events. Van Deventer et al. [25] recommended increased practice-based research to determine effective smoke risk communication strategies for government and media messaging. In particular, research is needed to assess the effectiveness, sufficiency, and reach of messages that include threat and efficacy information. Research is also needed to identify the audiences receiving government and media messages and to assess whether receiving these messages leads to behavioral change. The results are mixed on whether communities prefer local or government sources on wildfire and wildfire smoke messaging. Research is needed for any given geographic region, especially marginalized communities, to determine the most trusted and used channels for disaster and wildfire smoke communication. This review of literature demonstrated that trusted sources will differ depending on the community [25].

## 5. Conclusions

Improving familiarity with smoke risk reduction strategies and fire resources, including knowing where to turn, can be achieved with year-round educational campaigns across communities, potentially as part of weather forecasts. Accessible education efforts in multiple languages are needed to achieve protective public health goals in diverse and marginalized communities. Educational efforts and interactions should be initiated by local governments with special attention to marginalized and unincorporated communities. Although some research evidence exists for crafting effective alert and evacuation messaging, more evidence is needed on encouraging protective actions against repeat smoke exposure by emphasizing protective health benefits. Additionally, interagency coordination and consistency in messaging needs to be prioritized. Local government agencies involved in fire and air quality (smoke) need to strengthen routine communication with each other before engaging communities [18]. Regional air quality district agencies should provide smoke reporting systems accessible to the public and that the public can use to report local smoke events. More evidence is needed on protective action messaging against repeat smoke events threatening indoor air quality regardless of housing type. Additionally, employer policies and health protection messaging for outdoor laborers are needed. Finally, messaging on post-fire issues must be recognized including safe clean up and mental health, with gaps in the communication literature remaining.

## Figures and Tables

**Figure 1 ijerph-22-00368-f001:**
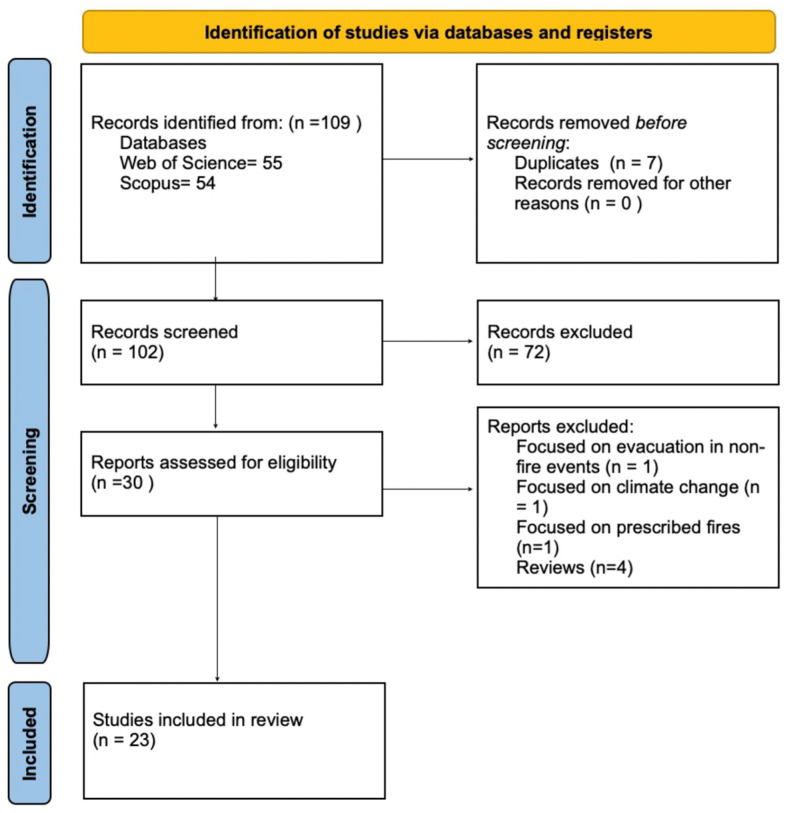
PRISMA flow chart [22].

**Table 1 ijerph-22-00368-t001:** Final 23 wildfire risk communication articles that met inclusion criteria.

Author	Year	Place	Population	Study Design
Hopfer et al. [18]	2024	Eastern Coachella Valley, California, USA	Vulnerable populations; low-income, non-English speakers, unincorporated communities	Focus groups
D’Evelyn et al. [24]	2023	Rural Washington, USA	Tribal and nontribal communities in Okanogan River Airshed Emphasis Area	Interview and focus group study to analyze perceptions of smoke exposure risk
Van Deventer et al. [25]	2021	Washington, USA	Residents of Okanogan, King, Pierce, Thurston, Yakima, Grant, Spokane, and Kittitas counties	Content analysis study of state wildfire website
Shellington et al. [26]	2022	British Columbia, Canada	Indigenous and non-Indigenous groups	Cross-sectional survey to identify how to improve public communication about wildfire smoke and health risks
Kirsch et al. [27]	2016	Bastrop County, Texas, USA	Residents	Survey study (longitudinal); Community Assessments for Public Health Emergency Response immediately after a wildfire and 3.5 years later
Rappold et al. [28]	2019	Washington and California, USA	Users of Online Smoke Sense App	Survey study
Hano et al. [29]	2020	USA	Users of Online Smoke Sense App	Survey study; app user perspectives on wildfire smoke as a health risk
Treves et al. [30]	2022	North Fair Oaks, California, USA	Vulnerable populations; low-income, non-English speakers	Interviews; practitioners at CACs
Wood et al. [31]	2022	Washington, USA	Indigenous people	Interview and focus group study
Aminpour et al. [32]	2022	Western states, USA	Online participants	Randomized message experiment
Walsh et al. [33]	2022	Australian Capital Territory and New South Wales, Australia	Youth	Focus group study
Marfori et al. [34]	2020	Huon Valley, Tasmania, Australia	Households	Focus group study
Slavik et al. [35]	2024	Oregon and Washington, USA	Social media users	Content analysis: institutional use of social media
VanderMolen et al. [36]	2024	Nevada, USA	Rural communities	Focus group study
Choy et al. [37]	2023	British Columbia, Canada	Canadian organizations and individuals	Survey study pre- and post-workshop
Hoshiko et al. [38]	2023	Mariposa County, California, USA	Medically vulnerable people	Cross-sectional survey
Thomas et al. [39]	2022	Victoria, Australia	Government officials (emergency risk communication professions)	Interview study
Errett et al. [40]	2019	Washington, USA	Practitioners and academics	Small group discussions at a symposium
Seale et al. [41]	2023	Australia	People with asthma or chronic obstructive pulmonary disease in bushfire-prone area	Semistructured phone interviews
Humphreys et al. [42]	2022	Washington, USA	Rural communities	Focus groups and interviews
Bice et al. [43]	2024	Colorado, USA	Outdoor workers	Survey
Cowie et al. [44]	2021	Australia	Experts on bushfire and representatives	Workshop panel discussion at a symposium
Dodd et al. [45]	2018	Northwest Territories, Canada	Indigenous community members	Semistructured in-person interviews

**Table 2 ijerph-22-00368-t002:** Twenty-three studies: lessons learned.

Author	Messaging Type	Individual or Community	Trusted Sources and Channels	Recommendations and Lessons Learned
Hopfer et al. [18]	Evacuation and smoke messaging	Community	Mix of traditional channels (television and radio) and nontraditional channels (printed alerts, sirens, and maps) Most trusted source: word-of-mouth or text communication with familiars and messaging with familiar community logos	Small unincorporated communities face lack of dynamic information during a wildfire or smoke emergency Residents expressed interest in fire-related training and developing an evacuation plan; Spanish materials needed Unincorporated communities experience outages, blocked roads, and food and water shortages during and after wildfires
D’Evelyn et al. [24]	Evacuation and smoke messaging	Community	Tribal and nontribal preferred local and community-based channels Facebook was most used to disseminate information Local sources trusted more than state or federal governments by both tribal and nontribal communities Trustworthiness determined by perceived credibility, quality of information, and relationship with the source	Clarify short-, medium-, and long-term impacts of wildfire smoke, including mental health impacts Clarify actionable steps Coordinate local groups Emphasize health Emphasize smoke-readiness Local sources are more trusted and, thus, more effective
Van Deventer et al. [25]	Evacuation and smoke messaging	Individual and community	Trusted source defined as messages that referenced a government agency or academic organization Government and media messages varied in type of trusted source	Greater coordination between government and news media about health risks of smoke exposure and vulnerable populations Use the same AQI across messengers to reduce confusion Year-round information campaign Local agencies should increase their role as trusted sources before and during smoke events Greater research on efficacy of government and media messaging Greater research about preferred channels and trusted sources
Shellington et al. [26]	Evacuation and smoke messaging	Individual and community	Diverse channels; mostly websites, social media, radio, and television	Disseminating tailored wildfire air quality smoke advisories via news outlets Year-round messaging Simplified messages with details linked online Employ diverse channels and in multiple languages
Kirsch et al. [27]	Evacuation and smoke messaging	Community	Different channels used for different types of information Trust in local government influenced by residents’ experience with exposure and past response efforts by local government	Greater trust in informal messaging via social media Greater trust in local organizations who have demonstrated community investment Address long-time distrust in larger government agencies People value expertise and transparency Political neutrality Information needs to be perceived as reliable, accurate, timely, and locally relevant Coordinate increased social media use throughout disaster management cycle Consider the source of information and its dissemination method
Rappold et al. [28]	Smoke messaging	Individual		Increase personal awareness of health risk instead of solely providing information about air quality Reinforce exposure-reduction behaviors Personalize messages and provide evidence that behavioral change is effective Reinforce benefits of protective health behavior
Hano et al. [29]	Evacuation and smoke messaging	Individual		Greater research in trusted sources
Treves et al. [30]	Smoke messaging	Community	Diverse channels including interactive mediums Practitioners of CAC considered outreach partnerships as trusted sources	Challenges of implementing CAC in city Limited at-home solutions provided Outreach to community was rushed and lacked resources Changing perception of preparedness compelled community to visit CAC Community suggested extended CAC hours, accessible transportation, and clear guidelines to identify CACs Need for increase in amenities, accessible transportation, resources, staffing needs, tools, and funding to disseminate information about CACs and smoke preparedness
Wood et al. [31]	Evacuation and smoke messaging	Community	Informal networks and social media Local sources trusted more than state or federal government by both tribal and nontribal communities Trustworthiness determined by perceived credibility, quality of information, and relationship with the source Perception of trust influenced by political ideology	Government agencies should demonstrate accountability to rural and tribal communities Build long-term relationships between government agencies and communities Include tribal nations in decision-making Center local perspectives and expertise in risk communication messaging Greater trust in local government Increase funding to support tribal partners in employment and resources Tribal and nontribal communities emphasized informal networks and private citizen-run fire watch Facebook groups
Aminpour et al. [32]	Smoke messaging	Individual	Facebook users preferred ads with narrative frame over ads with informational frame Facebook users had no preference between academic and government messengers	Minimal difference between preference for government (.gov) vs. academic (.edu) Preference for narrative ads over fact-focused ads Ads were effective and inexpensive, reaching large audience Use online platforms to conduct randomized messaging experiments
Walsh et al. [33]	Evacuation and smoke messaging	Individual	Diverse channels; mostly apps, social media, television, and radio Trusted source: World Health Organization	Design communication specific to age and developmental level of children Remind children of the presence of civic support to ensure appropriate behaviors during bushfires Kids focused on fire more than smoke risks Lack of kid-friendly, multilingual, multicultural resources Leverage culturally and linguistically diverse social networks to maximize outreach Diversify health messages
Marfori et al. [34]	Evacuation and smoke messaging	Individual	Diverse channels; Australian Broadcasting Corporation and social media were common; some advocated for a central resource Government agencies and community members were more trusted than social media	Protection messages from wildfires were contradictory or overshadowed smoke messaging American and Australian residents preferred television, radio, and communication with authorities, social networks, and local channels Most preferred simple messaging on social media Tailor messages about risks of different groups, long-term health harms, and risks associated with differing severities of smoke pollution Trusted central source Greater research to improve evidence for different health protection interventions Integrate smoke messages with wildfire messages Have a single website with links
Slavik et al. [35]	Evacuation and smoke messaging	Individual		Integrate protective motivation theory constructs in institutional messaging Combine numeric information, verbal cues, and AQI risk labels in communication Implement bidirectional communication Implement proactive messaging during smoke off-seasons Institutions used Twitter (X) to share smoke-related protective actions more than wildfire smoke risk information or community building Minimal tweets mentioned vulnerable populations and smoke mitigation measures Need for focus on vulnerable communities
VanderMolen et al. [36]	Smoke messaging	Community	County or local channels; greatest preference for government entities and community Facebook pages	Varied preference for channels but mostly preferred county website or social media page Least preferred communication was print flyers or pamphlets, most preferred was digital messaging Messaging should be inclusive to those with disability Knowledge of personal protective equipment was low; few knew the significance of using an N95 mask over a cotton or gauze mask Many interviewees reported not having access to a portable air purifier but did regularly check AQI for air quality reports
Choy et al. [37]	Evacuation and smoke messaging	Community		Workshops increased interest, participation, and knowledge of smoke communication Workshop participants reported interest in new smoke communication strategies for their organization Organizations reported main limitations: human resource capacity, legal barriers, and time pressure Create multistage and multiyear programs
Hoshiko et al. [38]	Evacuation and smoke messaging	Community	Diverse and local channels; differed depending on whether information focused on the past or future Trusted sources: county sheriff, county health agency, community, local fire departments	Clear action steps Provide preparation education via community events Promote air quality and smoke tracker apps (e.g., California Smoke Spotter) Deploy air resource advisors to wildfires to communicate smoke impacts as an additional form of resources and information messaging Rural areas: use mass phone calls, time-sensitive notifications, multiple channels, and noninternet options
Thomas et al. [39]	Evacuation and smoke messaging	Community		Outline what the government knows, what it does not know, and current actions for emergency messaging; gaps should be analyzed and addressed Emergency wildfire communication should explain that current advice is based on available information and subject to change; this motivates the audience to be alert and anticipate updates with new information Train practitioners on lived experience; create action plans Preparedness activities with fire drills or field exercises
Errett et al. [40]	Evacuation and smoke messaging	Community	Trusted channels and messengers can be identified using community-level asset mapping	Need better risk-related data Define hazard Greater research about subpopulations (e.g., values, behaviors, educational needs, and risk awareness) for consistent messaging Greater research on communicating risk to resistant or dismissive populations Greater research on how local governments address anxiety and fear Assess use of analogies and specific messages Consistencies across air quality indexes Evaluate effectiveness and reach of risk communications during previous wildfire smoke events
Seale et al. [41]	Evacuation and smoke messaging	Individual and community	Lack of trust in air quality apps Knowing information source builds trust Diverse channels including air-quality apps, websites, and social media groups	Mixed reviews about use of air-quality apps Healthcare professionals should encourage mask use Greater trust in knowing source of information Employ diverse channels including text alerts and automated calls Ensure people understand purpose of and steps to wearing masks Maximize communication efforts by learning from COVID-19 and other natural disasters
Humphreys et al. [42]	Evacuation and smoke messaging	Community	Diverse channels including push notifications on local apps Multiple distribution locations Multiple messengers	Implement information campaigns, expand AQI information, and use multiple formats Key informants were more positive about adequacy and coordination of communications than focus group participants; local and health and social service providers may have more wildfire smoke literacy Need for community-level access to information
Bice et al. [43]	Evacuation and smoke messaging	Individual and community	Diverse channels; mostly news outlets	Create messages that increase information seeking among outdoor workers Incorporate practical behavior recommendations in workplace messaging Address communication gap between advice for public vs. outdoor workers
Cowie et al. [44]	Evacuation and smoke messaging		Need for range of communication tools Difficult to construct relevant messages	Use different exposure guidelines including hourly reporting of air pollutant concentrations and air quality Reevaluate workplace standards and response Use lifespan-based messaging for healthy individuals Improve environmental health literacy Concern over media’s focus on crisis communication and misinformation Depoliticize dialogue on climate change Strengthen evidence through citizen science for policy decisions and the public
Dodd et al. [45]	Evacuation and smoke messaging	Community		Address different stages of prolonged wildfire and smoke event Incorporate smoke forecasting in communication Debrief after wildfire season for future planning Acknowledge community’s connection to the environment

## Data Availability

No new data were created in this systematic literature. Data sharing is not applicable to this systematic review. PRISMA methodology is reported in the Appendix A.

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
