# Peer review of "Wildfire and Smoke Risk Communication: A Systematic Literature Review from a Health Equity Focus"

_ijerph, 2025, doi:10.3390/ijerph22030368_

Round 1

Reviewer 1 Report

Comments and Suggestions for Authors

Review of Wildfire and smoke risk communication: a systematic literature review with a health equity focus

This study provides a systematic review of literature focused on wildfire and wildfire smoke risk communication strategies. I agree with the authors that this is a critical research area to focus on given the increasing frequency and severity of wildfires and smoke events. While the authors provide a detailed review of the available literature and identify key knowledge gaps, I believe the overall clarity and conciseness of the manuscript needs to be improved prior to publication. The topical focus of the review is unclear throughout the manuscript, and more details on the studies discussed need to be included. See below for specific comments.  

Comments:

1.    Since the review is focused on both wildfire and smoke risk communication, I would recommend mentioning smoke in the first few sentences of the abstract

2.    The sentence in the Abstract on lines 12-15 is quite confusing to read, as is the sentence on lines 18-19. I would recommend rewording these sentences to ensure they clearly communicate the correct message.

3.    Line 27: “… worsened smoke conditions post-fire as well” – severe smoke conditions can occur both during and after wildfires. I would recommend updating the sentence to reflect this.

4.    Lines 28-29: ”… as it is more toxic than other air pollutants” – while there is some epidemiologic evidence that the PM2.5 in smoke may be more toxic to respiratory endpoints, I would not say there is sufficient evidence to claim that wildfire smoke is “more toxic.” I would recommend updating this sentence to highlight that wildfire smoke may be more toxic to health than other air pollutants.

5.    Lines 31-32: there is not sufficient evidence to state that wildfire smoke causes asthma and COPD, rather most of the evidence shows that wildfire smoke can exacerbate these conditions and increase the risk of being hospitalized for these conditions. I would update this sentence to better reflect the evidence.

6.    In the first paragraph, I would recommend adding a sentence describing what wildfire smoke is and the mechanisms by which it affects health

7.    Line 38: Is the systematic review focused on communication of risks related to wildfire smoke, or risks related to wildfires and wildfire smoke? Based on the abstract, I assumed it was on both, but this paragraph makes it seem like that is not the case. I would update this sentence and paragraph to clarify the focus of the systematic review

8.    Lines 40-41: “These communities are often geographically isolated…” while I believe that this is true for some of the communities listed in the previous statement (i.e., rural, Indigenous, farm workers), it is not true for all communities listed (i.e., low income, elderly). I would clarify which communities you are referring to in this sentence. Also, could you add a reference for this sentence?

9.    Lines 43-45: Same as above – in this sentence, be explicit about which marginalized communities you are referring to

10. Lines 45-46: awareness and outreach are often not solely focused on saving lives, but also minimizing the acute and potentially severe health effects of wildfire smoke exposure. I would reword the sentence to clarify this.

11. Line 49: As with the comment pertaining to line 38, in this paragraph it would be good to clarify if the systematic review is focused on communication of risks related to wildfire smoke and wildfires, or just wildfire smoke

12. Lines 52-53: What about effective communication for acute smoke exposure? I’m assuming the review also addresses this

13. Line 63: It would be valuable to clarify here that this review is US-focused, to provide context that “international” refers to studies conducted outside of the US

14. Line 65: For the keywords, why was “Wildfire smoke” or “Smoke” not used as a keyword? Similarly, why was “risk” not used as a keyword? Terms related to smoke and risk seem critical to include in the search for relevant literature.

15. Lines 74-76: Please clarify what these two sentences mean – it is difficult to understand both of them in their current written form

16. In Figure 1, what do the asterixis refer to?

17. Lines 94-95: This sentence seems to indicate that the 4 reviews were included in the review, but Figure 1 seems to indicate that they were excluded. Please ensure that the text and Figure 1 are in agreement. Additionally, how is your systematic review different from the 4 reviews referenced here? It would be useful to add this information.

18. At the beginning of the results, it would be useful to provide an overview of geographic regions that the 17 studies took place in

19. In general, in the results, when describing specific studies, I would recommend adding details on where the study took place and the community of interest to provide sufficient context. i.e., Line 108, Line 114, Line 176, Line 231, Line 242, etc.

20. In general, in the results, I think it would be valuable to provide more information on the content of the communication – wildfire risk and wildfire smoke risk are different, as are health risk communication and evacuation communication, so it would be useful when describing the studies to be more specific about the type of communication the study was evaluating

21. Line 179: I would be explicit that you mean “adequate air filtration”

22. Line 203: “air quality smoke and health” – do you mean air quality or wildfire smoke here?

23. Line 215: Could you add details here on what type of websites?

24. The review is focused on communication related to wildfires and wildfire smoke, so I am wondering why papers on prescribed fires are included (e.g., Lines 224-225, Lines 261-267) – prescribed fires are quite different than wildfires, and communication surrounding prescribed fires seems to be outside the scope of this review. I’d recommend either updating the introduction to clarify that you’re also looking at prescribed fires and introduce what prescribed fires are, or remove the studies related to prescribed fires

25. Line 280: “Eight studies (n=8/17; 47%) focused on evacuation or smoke risk messaging.” – I thought that this was the key focus of the review, so if only 8 studies focused on this, it would be valuable to clarify what the other 9 studies focused on.

26. There are typos throughout the manuscript that should be addressed prior to publication (e.g., “Rappoled et al.” on Line 323)

27. Line 340: Please add detail about what the “Protection Motivation Theory” is

28. In Table 1, the “Place” and “Population” columns seem mixed up for some studies. For example, for Rappold et al., 2019, wouldn’t the population be Smoke Sense App users and the place be Washington and California? I would recommend revisiting this table to ensure it clearly communicates the studies’ details

29. In Table 2, I would recommend left justifying the last two columns – it is difficult to read when the bullet points are centered

30. Overall, reading the Discussion feels somewhat redundant with the Results – where possible, in the Discussion I would refrain from referencing the details of specific studies and focus on key summary points, take aways, and knowledge gaps.

Author Response

We thank the reviewers for the valuable time and helpful feedback to make this systematic review on wildfire communication more complete and clear. Thank you. Please see attached pdf for reviewer responses. 

Reviewer 2 Report

Comments and Suggestions for Authors

I appreciated reading your important and well done manuscript which will add to scientific and practice information.

I have only a few suggestions for changes:

1. In the Abstract you cite that your literature review covered until 2022, rather than 2024. However, in body of the manuscript, you mention end year as 2024, which is correct, given that you have literature cited between 2022 and 2024.

2. In the Methods, you describe well your inclusion on studies, however, there should also be mention of how you selected the key themes from those studies. This should include citation of the methodological process used, the number of research team members working on thematic coding, how differing opinions among team members were resolved and inter-rater reliability (if you tested that).

3. One other comment is mainly out of curiosity: in lines 328-334, there is mention of studies advocating for "personal tailoring" of communication.

"Messages should be targeted for different populations, especially for marginalized 326 populations [21, 28, 31]. As previously noted, health risk messages should be personally 327 tailored and detail the different risks associated with the degree of smoke severity[30, 34]. Messages should be targeted for different populations, especially for marginalized 326 populations [21, 28, 31]. As previously noted, health risk messages should be personally 327 tailored and detail the different risks associated with the degree of smoke severity[30, 34]. "

This might obviously seem like an impossible task for communicators to somehow know personal information about all the intended recipients and then tailor each message separately. I'd suggest double-checking if those authors really meant "personal tailoring" in this sense, or if they rather meant "group tailoring" for key segments of the population. If the authors did mean personal tailoring, it might be worth your putting in a comment about how difficult this might be.

4. There are a very few minor typos to correct.

Congratulations on a very good manuscript! I look forward to seeing it in print.

Author Response

We thank the reviewer for their valuable time and feedback! Please see attached responses. Thank you again for making this systematic review on wildfire communication more complete and clear. 

Round 2

Reviewer 1 Report

Comments and Suggestions for Authors

Thanks to the authors for their in-depth revision of the manuscript and response to the reviewer comments - all of my major concerns have been addressed and I think the manuscript is well-written. I have a few minor comments that could be addressed prior to publication:

Line 50: The increased risk of hospitalizations / ED visits associated with wildfire smoke exposure is considered an acute effect (since hospitalization / ED visits are an acute outcome), not a long-term effect. It might be valuable to clarify this, since we actually know very little about the long-term health risks of wildfire smoke. 

Line 110: "Sociological Abstracts was excluded" - it would be valuable to clarify what you mean by 'Sociological Abstracts'

Figure 1. Is the "+" symbol next to the "Screening" label supposed to be there? It might be good to clarify in the caption that the flow chart represents the initial search round (e.g., “wildfire” AND “communication” AND “public health" leading to 109 results

Author Response

 Dear Editor of IJERPH, 

We have addressed the three minor revisions requested by the reviewer. We changed line 50 to frame the increased risk of hospitalization with wildfire smoke risk as acute (not long term). Line 110, we clarified that the database Sociological Abstracts was excluded in the search process as it did not yield additional unique articles. Finally, Figure 1 (PRISMA flowchart of search process) was clean up and we removed an erroneous + . We thank the reviewers again for their time and feedback!

Sincerely, 

Suellen Hopfer

Public health professor and corresponding author